# Associations between Maternal Dietary Patterns, Biomarkers and Delivery Outcomes in Healthy Singleton Pregnancies: Multicenter Italian GIFt Study

**DOI:** 10.3390/nu14173631

**Published:** 2022-09-02

**Authors:** Gaia Maria Anelli, Francesca Parisi, Laura Sarno, Ottavia Fornaciari, Annunziata Carlea, Chiara Coco, Matteo Della Porta, Nunzia Mollo, Paola Maria Villa, Maurizio Guida, Roberta Cazzola, Ersilia Troiano, Monica Pasotti, Graziella Volpi, Laura Vetrani, Manuela Maione, Irene Cetin

**Affiliations:** 1Department of Biomedical and Clinical Sciences, Università degli Studi di Milano, 20157 Milan, Italy; 2Department of Woman, Mother and Child, Luigi Sacco and Vittore Buzzi Children Hospitals, ASST Fatebenefratelli-Sacco, Università degli Studi di Milano, 20154 Milan, Italy; 3Department of Neurosciences, Reproductive Sciences and Dentistry, Università degli Studi di Napoli Federico II, 80131 Naples, Italy; 4Department of Molecular Medicine and Medical Biotechnology, Università degli Studi di Napoli Federico II, 80131 Naples, Italy; 5Nutrition and Dietetics Technical Scientific Association (ASAND), 95128 Catania, Italy

**Keywords:** GIFt study, maternal nutrition, maternal dietary patterns, RBCs folate, vitamin D, hepcidin

## Abstract

*Background:* Maternal nutrition represents a critical risk factor for adverse health outcomes in both mother and offspring. We aimed to investigate associations between maternal nutritional habits, biomarker status, and pregnancy outcome among Italian healthy normal-weight pregnancies. *Methods:* Multicenter prospective cohort study recruiting Italian healthy normal-weight women with singleton spontaneous pregnancies at 20 ± 2 weeks (T1) in Milan and Naples. All patients underwent nutritional evaluations by our collecting a 7-day weighed dietary record at 25 ± 1 weeks (T2) and a Food Frequency Questionnaire at 29 ± 2 weeks (T3). Maternal venous blood samples were collected at T3 to assess nutritional, inflammatory and oxidative biomarker concentrations (RBCs folate, vitamin D, hepcidin, total antioxidant capacity). Pregnancy outcomes were collected at delivery (T4). General linear models adjusted for confounding factors were estimated to investigate associations between maternal dietary pattern adherence, nutrient intakes, biomarker concentrations and delivery outcomes. *Results:* 219 healthy normal-weight pregnant women were enrolled. Vitamin D and RBCs folate concentrations, as well as micronutrient intakes, were consistently below the recommended range. In a multi-adjusted model, maternal adherence to the most prevalent ‘high meat, animal fats, grains’ dietary pattern was positively associated with hepcidin concentrations and negatively associated with gestational age at delivery in pregnancies carrying female fetuses. Hepcidin plasma levels were further negatively associated to placental weight, whereas vitamin D concentrations were positively associated to neonatal weight. *Conclusions:* A high adherence to an unbalanced ‘high meat, animal fats, grains’ pattern was detected among Italian normal-weight low-risk pregnancies, further associated with maternal pro-inflammatory status and gestational age at delivery. This evidence underlines the need for a dedicated nutritional counseling even among low-risk pregnancies.

## 1. Introduction

Maternal nutrition is the main determinant of fetal nutrition and represents a critical and modifiable risk factor for adverse pregnancy and long-term health outcomes in both mother and offspring [1,2,3]. Qualitative malnutrition (i.e., excessive macronutrient and energy with concurrent inadequate micronutrient intake) is spreading in reproductive age women in both low- and high-income countries, as a result of the increasing adherence to high-energy, low-cost and fast-food dietary patterns [4,5]. Nutritional deficiencies have been associated with altered oocyte quality, maternal pro-inflammatory status and maladaptation to pregnancy, finally resulting in deranged embryogenesis, placental dysfunction, and poor pregnancy outcomes [6,7,8]. In contrast, a high adherence to the Mediterranean/prudent diet is associated to increased reproductive success, improved embryonic growth and fetal development, with protective effects against metabolic syndrome in later life [9,10,11].

Despite these data, most couples planning or entering pregnancy show qualitative malnutrition, with a 47% prevalence of unhealthy periconceptional body mass index (BMI) in the Italian female population (Istat 2012 data) [4,12,13].

The assessment of maternal nutritional habits and status should be therefore crucial in standard pregnancy care to evaluate deficiencies and ensure optimal exposure. A dedicated nutritional counselling, and the use of validated and reproducible tools aimed at identifying dietary pattern adherence, may help clinicians to detect maternal malnutrition and prevent adverse effects [14,15,16]. To date, no conclusive data are available on prevalent dietary patterns and maternal biomarker status in Italian healthy pregnancies [17]. Although several studies investigated the association between qualitative malnutrition and pregnancy outcomes, no decisive data are available on the effect of nutritional counselling among low-risk pregnancies, in contrast to more corroborated effects of dietary counselling supplementation among high-risk pregnancies (i.e., gestational diabetes, hypertension, obesity) and low-income countries [18,19].

The multicenter prospective hospital-based cohort study named GIFt (‘Gestational Intake of Food towards healthy outcomes’) aimed to investigate the associations between maternal nutritional habits, biomarker status of systemic inflammation and oxidative stress, and pregnancy outcomes among Italian healthy normal-weight pregnancies.

Two geographical areas were included as representative of a Continental (Milan) and Mediterranean (Naples) area.

The study had the following main objectives:Collect energy and nutrient intakes by 7-Day weighted dietary Record (7-DR), and compare with Italian nutritional recommendations;Define prevalent dietary patterns in a population of healthy normal-weight Italian pregnancies as extracted from Food Frequency Questionnaires (FFQ);Characterize maternal oxidative and inflammatory status by quantifying blood biomarkers;Investigate associations between maternal nutritional intakes and dietary patterns, blood biomarkers of inflammation and oxidative stress, and pregnancy outcomes, also stratifying for geographical area.

## 2. Materials and Methods

### 2.1. Study Protocol

The GIFt study is a multicenter prospective hospital-based cohort study conducted at the Departments of Obstetrics and Gynecology of ‘Luigi Sacco’ (Milan), ‘Vittore Buzzi’ (Milan) and ‘Federico II’ Hospitals (Naples), Italy.

The study was conducted in accordance with the Declaration of Helsinki and in compliance with all current Good Clinical Practice guidelines, local laws, regulations and organizations. The protocol was approved by the “Medical Ethical and Institutional Review Board”; all participants gave their informed consent to collect personal data and biological samples (Institutional Review Board ‘Comitato Etico Milano Area 1’ Prot. n° 8292/2016 and 33043/2018).

### 2.2. Study Participants

Healthy normal-weight singleton pregnancies at 20 ± 2 gestational weeks (baseline, T1) were eligible for participation and consecutively recruited between January 2017 and June 2020 at the Obstetric Units of ‘Luigi Sacco’, ‘Vittore Buzzi Children’ (Milan) and ‘Federico II’ (Naples) Hospitals.

Only Italian healthy women with singleton spontaneous pregnancies, aged between 20–40 years, with a pre-gestational Body Mass Index (BMI) between 18.5 and 24.9 kg/m^2^ [20] and a basal glycemia < 92 mg/dL at enrollment were eligible for recruitment. Exclusion criteria: any known maternal pregestational disease or chronic therapy, substance or alcohol abuse, and smoking habit (up to 5 cigarettes per day until the 6th gestational week). Women following a specific diet (e.g., celiac disease, vegetarian diet, vegetarian diet) were excluded according to the study focus. Further exclusion or dropout criteria: congenital or chromosomal anomalies, insurgence of pregnancy complications (e.g., preeclampsia, gestational diabetes).

### 2.3. Study Design and Data Collection

The study design included four steps (T1, T2, T3 and T4), as summarized in Figure 1.

At enrollment (T1), all participants filled out a general questionnaire detailing demographic, obstetric and clinical data (age, educational level, employment and marital status, obstetric and medical history, vitamin supplement use). Blood pressure and anthropometric (height, weight) measurements were obtained by trained counselors and self-reported pre-pregnancy weight for pregestational BMI and Gestational Weight Gain (GWG) calculation was recorded. First trimester basal glycemia and hematological status, including hemoglobin, hematocrit and medium corpuscular volume (MCV), were measured. Delivery outcomes, including gestational age, mode of delivery, pregnancy complications, placental weight and neonatal data (weight, sex, length, head circumference and Apgar score) were recorded at delivery (T4).

#### 2.3.1. Nutritional Data Collection

As defined in Figure 1, nutritional data were collected by using the 7-day weighed dietary record (7-DR) and the Food Frequency Questionnaire (FFQ). In order to assess energy, i.e., macro- and micronutrient intakes, all women were asked to fulfill a 7-DR at T2 (25 ± 1 gestational week-gw) within the II trimester. Supported by a trained dietitian and contextually informed about food consumption, each woman recorded the specific type of food, drink and/or seasoning on a provided format for 7 consecutive days (equivalent to 26 ± 1 gw), after which the diaries were returned. The raw or net food weight/portion were recorded. To obtain the highest accuracy in filling in 7-DRs, the dietitians adequately trained the survey participants in describing the foods, amounts consumed, cooking methods, etc. At the end of the recording period, the 7-DR was thoroughly reviewed with the subject.

The 7-DRs were then entered into a specific Microsoft Excel spreadsheet (Microsoft Corporation, Redmond, WA, USA) by trained dietitians. Each frequency consumption was reported as a factor, derived from the ratio of the frequency per day, week and month each food item was consumed, in order to obtain a daily fraction of consumption for each item. A dietary bromatological analysis was performed on the obtained data, as previously reported [21]. Macro- and micronutrient intakes were assessed by us using Food Composition Database for Epidemiological Studies in Italy (Banca Dati di Composizione degli Alimenti per Studi Epidemiologici in Italia—BDA) [22].

By comparing the energy and nutrient intake of the study population to the Italian references [23], we estimated the reference levels of energy and protein intake (Energy Intake for Population (EIP) and Protein Intake for Population (PIP)). Based on LARN for the non-pregnant female population aged 30–59 years, we chose a height average of 160 cm (proper of our study population) and a weight average of 60 kg (described as standard weight for an Italian woman). Concerning Physical Activity Level (PAL), we considered the 3 main levels described by LARN (1.45 for sedentary activity, 1.6 for moderate activity and 1.75 for elevated activity) as well as their average, excluding the highest PAL of 2.1, which was inconsistent with our population possible PAL. To obtain EIP and PIP, we added up the estimated mean energy (EIF: +266 kCal for second trimester) and protein cost (PIF: +8 g for second trimester) of pregnancy according to LARN.

Maternal dietary intake and pattern adherence were assessed at T3 using the validated semi-quantitative food frequency questionnaire (FFQ) developed for the Italian population [24], with minor adjustments based on the latest food consumption and tastes. The nutritional data recorded by this FFQ concerned the preceding three months. This temporal choice depended on the need that both data collected from 7-DRs and FFQs were related to the same pregnancy trimester (specifically II trimester). Indeed, a future aim of this protocol will be to validate the FFQ used in this study cohort.

The FFQ consists of 192 food items structured according to meal patterns and includes questions on consumption frequency, portion size, and preparation method. The FFQs were finally checked for completeness and consistency by a trained dietitian and unreliable energy reporting were excluded from final analyses. We reduced 192 food items to 15 predefined food groups based on similar origin and nutrient content (dairy, cereals, grains, vegetables, legumes, potatoes, meat, fish, egg, fruit, nuts, vegetable fats, animal fats, sweets and non-alcohol beverages).

#### 2.3.2. Blood Testing: Nutritional, Inflammatory and Oxidative Biomarkers

Maternal venous blood was collected from a radial vein at T3.

*Heparin and EDTA Blood Processing:* samples were centrifuged at 1000× *g* × 10′ at 4 °C to separate plasma and corpuscular components; plasma was stored at −80 °C. The corpuscular phase was then resuspended in 0.2 M EDTA + 150 nM NaCl solution in order to wash off any possible residual plasma. Samples were then centrifuged at 2000× *g* × 15′ at 4 °C, and stored at −80 °C.

*Gel Acrylic Blood Processing:* samples were centrifuged at 2000× *g* × 10′ at RT to obtain serum, then stored at −80 °C.

A ‘folate microbiological assay’ was performed to quantify maternal folate levels in red blood cells (RBCs), following the guidelines by CDC NHANES 2011-2012 [25]. As previously reported [26,27], the experimental protocol was revised to be adapted to the erythrocytes, as described by Piyathilake and colleagues [28]. Briefly, red blood cells samples were unfrosted by their placing in a thermostatically controlled water bath (5′ at 37 °C), then diluted thirty-times in 1% ascorbic acid. Diluted-packed erythrocytes’ hemolysates were then added to an assay medium containing Lactobacillus Rhamnosus (ATCC 27773, NCIB 10463 by American Type Colture Collection-ATCC, Manassas, VA, USA) plus all nutrients needed for its growth, except for folate. The inoculated medium was incubated for 45 h at 37 °C. L. Rhamnosus’ growth progressively increases the turbidity of its medium, which is directly proportional to the folate amount. Folate levels were therefore measured in the inoculated medium at 531 nm with a microplate reader (Victor V 1420, Perkin Elmer Italia S.p.A, Milan, Italy). The measure was calibrated with 5-methyltetrahydrofolic acid (5MeTHF). The chemicals used for the assay were the following: ascorbic acid, sodium ascorbate, folic acid medium, chloramphenicol, tween-80, manganese sulfate, sodium azide, glycerol (all chemicals by Merck, Milan, Italy) and human serum folate free (Trina Bioreactives, Nänikon, Switzerland). The results were expressed as ng/mL according to [25].

The quantitative determination of total vitamin D in serum was made by a competitive chemiluminescence method performed on the Liaison XL platform using the 25-OH vitamin D Assay (Liaison 25-OH Vitamin D Total Assay Clia, DiaSorin S.p.A, Vercelli, Piemonte, Italy). Details are available from the manufacturer’s instructions [29].

Hepcidin (Hpn-25) concentration was measured in plasma by a competitive solid-phase ELISA (EIA-4705, DRG Diagnostics; Marburg, Germany), as previously described in [30].

The measurement interval was [1.09–13.72 ng/mL]. Hpn-25 ELISA limit of detection was 0.304 ng/mL and the intra- and inter- assay coefficients of variation (CV%) were both 5%.

Total Antioxidant Capacity was measured in maternal heparin plasma with an antioxidant assay kit (Antioxidant Assay Kit- Item 709001; Cayman Chemicals, Ann Arbor, MI, USA). This method detects both aqueous- and lipid-soluble antioxidants, thus assessing the combined antioxidant activities of all its constituents including proteins (e.g., glutathione), lipids, vitamins (e.g., Vitamin C and E) or uric acid [31]. Samples were diluted 1:30 and analyzed in duplicate, following the kit-enclosed instructions. The absorbance was detected at 750 nm.

The measurement interval was [0.332–6.470 mM]; the intra- and inter- assay coefficients of variation (CV%) were respectively 3% and 9%.

### 2.4. Statistical Analysis

Maternal characteristics, biomarker concentrations, nutritional intake from 7-DR and delivery outcomes were compared between subgroups (Milan compared to Naples) by using the Chi-square or exact tests for ordinal variables, and Mann–Whitney U test or Student’s *t*-test for continuous variables. Vitamin D seasonality was compared among groups by the Kruskal–Wallis test, with post hoc analyses by Mann–Whitney U test (with Bonferroni correction for multiple groups: *p* ≤ 0.017).

Dietary patterns were extracted from reliable FFQs by using the Principal Component Analysis (PCA). This is a standard multivariate statistical technique that aggregates specific food groups into complex dietary patterns according to the degree to which food items are reciprocally correlated, as detailed in [11,32]. Only the first three dietary patterns (i.e., principal components) with eigenvalues ≥ 1.1 were extracted, thus reducing the bias of multiple testing. After performing the PCA, two values were automatically assigned to food items (factor loadings) and women (component scores). Each food item was firstly assigned a factor loading, indicating how much it correlated to the extracted dietary pattern (positive factor loadings indicated positive correlation to the dietary pattern and vice versa, whereas the absolute value indicated the strength of correlation). We used three factor loadings, with the highest absolute value for the dietary pattern labelling. Additionally, every woman was automatically assigned a component score representing her adherence to the extracted dietary pattern (again, the higher the component score, the greater the adherence of the woman to that specific dietary pattern). Dietary pattern (component) adherence (scores) was compared between the two study subgroups by using Mann–Whitney U test. To estimate associations between maternal dietary pattern adherence (component scores), nutrient intakes (7-DR), biomarker concentrations, and delivery outcomes, general linear models adjusted for confounding factors were estimated. A log10 transformation of non-normally distributed variables was firstly performed to approximate Gaussian distributions. Confounding factors were chosen based on the previous literature and significant differences between the study subgroups (maternal age, pregestational BMI, education, working status, parity, folic acid/multivitamin supplement use).

*p*-values < 0.05 were considered statistically significant. All analyses were performed by using the statistical package SPSS, v.27 (IBM; Armonk, NY, USA).

## 3. Results

A total of 219 healthy normal-weight pregnant women were enrolled in Milan (*n* = 112) and Naples (*n* = 107). Screening failure (*n* = 21) and drop-out (*n* = 19) were equally distributed in the two subgroups, with the latter due to maternal choice or insurgence of other pregnancy complications (e.g., gestational diabetes, infections, congenital/genetic abnormalities).

Maternal baseline characteristics with comparison between the two study subgroups are reported in Table 1.

Pregnant women from Milan were more frequently workers, not married, and supplemented with multivitamins compared to women from Naples, whereas no differences were detected in maternal age and pregestational BMI. According to the inclusion criteria, the latter ranked in the normal weight range by definition. The analysis of biomarker concentrations at T3 showed RBCs folate concentrations significantly higher in Naples compared to Milan, with mean values of the total study population consistently lower than the recommended safe value in pregnancy (906 ng/mL). Despite being representative of Continental versus Mediterranean areas, no differences were detected in serum vitamin D levels in the two study subgroups, with mean values depicting a status of vitamin D deficiency. Seasonal variations of serum vitamin D were also tested (*n* = 81 from Milan, *n* = 85 from Naples subjects), showing higher concentrations in autumn (23.5 ± 6.3, *p* = 0.01; Milan: *n* = 20, Naples: *n* = 22) and summer (25.8 ± 8.7, *p* = 0.02; Milan: *n* = 13, Naples: *n* = 25) compared to winter (19.31 ± 6.13 μg/L; Milan: *n* = 31, Naples: *n* = 31).

Hepcidin (Hpn) plasma concentrations, detected as Hpn-25 aa mature peptide, were significantly lower in Naples compared to the Milan subgroup, but no attested reference values in the pregnant women population have been defined so far.

Total antioxidant capacity was similar in pregnant women of the compared groups.

No differences in hematocrit and hemoglobin concentrations were recorded in the two study subgroups, while serum ferritin was significantly lower in Naples’ subgroup (Mann–Whitney U test: *p* = 0.04).

Table 2 lists pregnancy outcome at T4.

Significant differences between the subgroups were detected for GWG at term, gestational age at delivery, and neonatal head circumference. Significant higher rates of cesarean section were recorded in Naples compared to Milan (chi-square test: χ^2^ (1, *n* = 171) = 4.512, *p* = 0.034, ϕ = 0.176). Neonates were equally distributed in term of sex (M: *n* = 73, F: *n* = 98) between Milan (M: 37.1%, F: 62.9%) and Naples (M: 47.8%, F: 52.2%).

A low incidence of remarkable pregnancy complications was recorded (dropouts), with only three women diagnosed with gestational diabetes (1.7%), one hypertensive disorder and no preterm delivery. No differences according to the growth and development curves were found in the frequencies of AGA (Appropriate for Gestational Age, *n* = 132, 79.5%), SGA (Small for Gestational Age, *n* = 24, 14.5%) and LGA (Large for Gestational Age, *n* = 10, 6.0%) neonates, independently of geographical area.

Macro- and micronutrient intakes were comparable depending on geographical area. The mean energy intake was also comparable between Milan (2027.8 kcal/die) and Naples (1694.5 kcal/die) subgroups, but lower than LARN recommendations (2362 kcal/die). In contrast, protein intake was higher (75.9 g in the total study population) than the recommended (62 g/die). All pregnant women failed in reaching the recommended intakes of cholesterol (<300 mg/die), carbohydrates (45–60%) and soluble carbohydrates (<15%), since the declared consumptions were respectively 249.4 mg/die (equal to 83.1%), 70.5% and 85.9% compared to LARN. Even the fibers consumption (21.8 gr/die) ranked below the recommended intake (25 gr/die). High intakes of sodium (3459.8 mg/die) compared to LARN (1500 mg/die) were also reported. Conversely, iron (11.3 mg/die), calcium (73.3 mg/die), iodine (83.8 μg/die) and folate (312.9 μg/die) intakes were severely deficient compared to LARN recommendations (27 mg/die, 1200 mg/die, 200 μg/die, 600 μg/die respectively).

Figure 2 shows energy and nutritional intakes expressed as percentage of LARN recommendations.

Three dietary patterns were obtained by using the PCA from the FFQ analysis, explaining 33.4% of the overall maternal dietary intake variance in the total study population (Table 3).

Based on the highest absolute value of factor loadings, the first component was labeled as ‘high meat, animal fats, grains’ dietary pattern, thus resembling a ‘Western’ dietary pattern. The second component was labeled as ‘high fish, fruits, nuts’ and resembled a ‘Mediterranean’ dietary pattern. No differences in the adherence to these dietary patterns were detected between the two study subgroups. The third component was highly associated with egg and sweets consumption, with the subgroup of Naples showing a higher adherence compared to Milan (Mann–Whitney U test: *p* = 0.01).

Table 4 shows the results from the general linear models adjusted for confounding factors as previously defined.

No associations were detected between ‘high fish, fruits, nuts’, ‘egg, sweets and low legumes’ dietary patterns and biomarker status or delivery outcomes (data not shown). Conversely, maternal adherence to the ‘high meat, animal fats, grains’ dietary pattern was positively associated with Hpn-25 concentrations and negatively associated with vitamin D concentrations at T3. Furthermore, the same dietary pattern showed a negative association with gestational age at delivery only in pregnancies carrying female fetuses, meaning lower gestational age at delivery in women highly adherent to the ‘high meat, animal fats, grains’ dietary pattern. No associations were detected between dietary pattern adherence and other delivery outcomes. When single food group or macro/micro- nutrient intakes were taken into account, no significant associations were recorded with either delivery outcomes or maternal biomarkers (data not shown).

Concerning maternal biomarkers, Hpn-25 plasma levels at T3 were negatively associated with placental weight in the total study population, whereas vitamin D concentrations were positively associated with neonatal weight. The analysis stratified for geographical subgroups did not substantially change the detected associations (data not shown).

## 4. Discussion

The multicenter longitudinal GIFt describes maternal nutritional habits, providing detailed dietary pattern adherence and nutrient intakes, and inflammatory/oxidative status in a highly selected and homogeneous population of Italian healthy normal-weight pregnant women from two geographical contexts, as representative of Continental and Mediterranean areas. The study population was homogenous for almost all baseline features, these being maternal age, pregestational BMI and educational level comparable between the study subgroups. All included women were a normal weight by definition and with a GWG within the proper IOM ranges, despite a higher GWG at term in the Milan subgroup. A low incidence of pregnancy complications was detected, as expected due to the very strict inclusion criteria. The higher cesarean section rate recorded in Naples was in line with Italian epidemiological data [34].

In this context of physiological, normal-weight and strictly defined healthy pregnancies, the results from the nutritional analyses seem more relevant. Nutrient intakes from 7-DRs showed a worrying distance from recommendations, despite the reported energy intake being lower than the recommended in both total study population and subgroups. In line with this result, we recently recorded energy intakes lower than recommended in a cohort of Italian normal- and overweight pregnant women [21]. The Mediterranean PHIME cohort additionally confirmed this result and the authors suggested that pregnant women did not show any energy deficit, probably due to the combination of inadequate eating habits and low physical activity level [10].

The present study further showed an excessive intake of proteins in the study population, whereas all pregnant women failed in reaching the recommended intakes of fibers, cholesterol, carbohydrates and soluble carbohydrates, again resembling the previously reported dietary deficiencies [25,35,36].

Among micronutrients, higher intakes of sodium and phosphorus were opposed to extremely deficient intakes of iron, calcium, iodine and folate compared to the LARN recommendations. Data from the Norwegian MoBa Cohort study reported higher intakes of sodium in pregnant women with inadequate consumption of organic food and fiber-rich diet [36]. Low dietary intakes of iron, calcium, iodine and folate have been increasingly reported in pregnant women in both European and non-European countries [37].

Folates are critical for several biochemical reactions (e.g., amino acids production, DNA synthesis and methylation, homocysteine metabolism and nitric oxide production). A folate deficiency causes an increase of inflammatory cytokines’ release, oxidative stress and apoptosis. The growing adherence to a ‘Western resembling’ diet makes difficult a proper intake of food-derived folate from fruits, vegetables, and cereals, thus increasing the risk of maternal hyperhomocysteinemia and of neonatal spina bifida or other neural tube defects.

All these results are in line with the present FFQ analysis. Unexpectedly, the most prevalent dietary pattern in a sample of healthy normal-weight Italian pregnancies was the ‘high meat, animal fats, grains’ DP, highly correlated with protein, fats and sodium intake, and consistently resembling a Western-type dietary pattern. A recent Italian study confirmed a similar higher adherence to a ‘Western resembling’ pattern in the third trimester in women with increased GWG [38].

Importantly, the ‘high meat, animal fats, grains’ dietary pattern was the only dietary pattern further associated with maternal biomarker status at T3, as well as with gestational age at delivery in pregnancies carrying female fetuses in multi-adjusted models considering confounding factors. In detail, the ‘high meat, animal fats, grains’ dietary pattern was negatively associated with serum vitamin D and positively associated with hepcidin plasma concentrations. A rising interest for predictive inflammatory biomarkers and their associations with maternal nutrition has developed. A recent systematic review reported how maternal dietary patterns characterized by higher intakes of animal protein and cholesterol and/or lower intake of fiber were associated among the other factors with pro-inflammatory markers, including CRP, IL-6, TNFα, IL-8 [17].

We chose hepcidin as relevant biomarker for its double role of inflammatory protein and negative regulator of iron homeostasis. No reference ranges are available, despite concentrations being known to decrease throughout pregnancy [39]. Compared to previous data on obese pregnancies, here we report lower concentrations than expected in lean healthy pregnancies [30]. Furthermore, we detected a positive association between a ‘Western resembling’ dietary pattern and maternal hepcidin in a healthy normal-weight population, possibly explained by the pro-inflammatory characteristics of the extracted dietary pattern with increased intakes of animal protein and fats. Interestingly, hepcidin plasma concentrations were also negatively associated to placental weight, suggesting hepcidin as a potential predictor of placental development and function.

Maternal vitamin D concentrations highlighted an hypovitaminosis state in the study population and subgroups, further explained by the negative association between the most prevalent ‘high meat, animal fats, grains’ dietary pattern and the biomarker concentrations. Despite the availability of suitable nutritional foods and supplements and a normal pregestational BMI, Italian pregnant women follow an imbalanced diet with inadequate intakes of folate, vitamin D, iron and iodine, finally reflected in altered biomarker status and delivery outcomes as shown in this population [40]. Severe vitamin D deficiency [<20–30 ng/mL] has been related to both maternal—e.g., hypertensive disorders, gestational diabetes and risk of caesarean section—and neonatal adverse outcomes [37,41,42]. Similarly, we detected a positive association between vitamin D concentrations and neonatal weight, the main proxy for adequate intrauterine growth and development. Seasonality of vitamin D was expected, considering that sunlight (UV-B) is the key to its subcutaneously synthesis. The disappointing data were the serum vitamin D, comparable between Milan and Naples. This agrees with evidence from others Mediterranean countries, where the high levels of sunshine collide with a poor dietary vitamin D intake (e.g., oily fish or fortified dairy products), reduced sunshine exposure and inappropriate health policies [43,44].

No other associations were detected between maternal dietary pattern adherence and studied biomarkers. Furthermore, no single food item or nutrient intake showed relevant association with maternal biomarker status, indicating the importance of considering complex dietary patterns as potent exposures capable to impact health status and outcome [45].

Lastly, inverse associations were detected between maternal adherence to the ‘high meat, animal fats, grains’ dietary pattern and gestational age at delivery in a subgroup of female-carrying pregnancies. Despite being statistically significant, we detected a 4-day difference in gestational age at delivery between women highly vs. poorly adherent to the ‘high meat, animal fats, grains’ dietary pattern, which might be clinically irrelevant at term. Nevertheless, as this result was obtained in healthy singleton normal-weight uncomplicated pregnancies, it is possible that by extending the inclusion criteria to high-risk pregnancies (e.g., obesity, preterm deliveries) the detected effect might be more clinically relevant. Indeed, recent data are available about sexual dimorphism on early exposure to maternal high-fat diet (HFD) in perinatal period. Gemici and colleagues highlighted signs of neuroplastic changes stemming from autonomic circuits imbalance with increased sympathetic tone in the early adulthood of male offspring rats [46]. The triglyceride profile varies between adipose depots in mice dam fed with HFD depending on sexes, with increased thermogenesis and cell differentiation in BAT only in female offspring [47]. It will be promising to develop this initial evidence from animal studies in order to understand how maternal diet could influence intrauterine life in a sex-specific manner.

As already discussed, the same dietary pattern impacted on maternal inflammatory status, as indicated by increased hepcidin concentrations. We may hypothesize that the nutritional chronic insult deriving from an unbalanced dietary pattern might exacerbate a proinflammatory status (increased Hpn-25 concentrations) which finally contributes to decreased placental weight and gestational age at delivery, even in healthy normal-weight pregnancies [48]. The sex-specific strategies to cope with maternal detrimental exposures are already known [49,50]. Transcriptome differences, sex-specific epigenetic mechanisms and dimorphic placental responses may explain different adaptive responses to nutritional insults, finally resulting in sex-specific fetal growth and placental effects [51,52].

### Strengths and Limitations

The main strength of the GIFt study is the strictly defined and highly homogeneous population of healthy, normal-weight women, carrying singleton spontaneous physiological pregnancies. This represents a very low-risk population, as a starting reference point for future evaluations of high-risk pregnancies. In fact, we definitely expect additional associations when including maternal obesity or pregestational chronic disease, as preexisting inflammatory conditions possibly deranged by unbalanced dietary patterns and pregnancy adaptations. Secondly, the study design involved longitudinal nutritional assessments throughout validated tools (7-DR and FFQ) performed by trained dietitians and providing an accurate picture of maternal nutritional habits. The multicenter design additionally provided the possibility to evaluate geographical areas with a potentially different nutritional, cultural and clinical approach. Maternal systemic biomarkers, in particular serum vitamin D, RBCs folate and plasma hepcidin, were detected and quantified by ‘gold-standard’ experimental methods.

Conversely, several limitations need to be pointed out. Firstly, the reported associations cannot be generalized to the almost more represented obstetric population of overweight/obese or high-risk pregnancies. In addition, this study focused on the second half of pregnancy (T1 at 20 weeks), thus excluding the most relevant period where fertilization, implantation, embryogenesis, and placentation take place.

No information on physical activity was recorded, which notoriously impacts on maternal inflammatory and metabolic status, as well as on delivery outcomes. No additional inflammatory biomarkers have been included (e.g., cytokines, CRP).

Finally, a definitely low energy intake reporting and percentage of pregnancy complications (e.g., no preterm deliveries, one gestational hypertension) has been detected in the study population. Despite the highly selected healthy population being possibly able explain the result, we cannot exclude the possibility that spontaneous maternal modifications of nutritional habits towards a healthier behavior were realized. In fact, repeated investigations of nutritional habits through 7-DRs and FFQs might have created individual consciousness, despite no nutritional counselling or intervention was performed.

## 5. Conclusions

This study describes the nutritional habits of healthy normal-weight pregnant women belonging to a Continental (Milan) compared to Mediterranean area in Italy. This population shows a high adherence to an unbalanced ‘high meat, animal fats, grains’ pattern, further associated with maternal pro-inflammatory status and gestational age at delivery. These results underline the crucial need for a detailed and dedicated nutritional counseling even among low-risk pregnancies.

## Figures and Tables

**Figure 1 nutrients-14-03631-f001:**
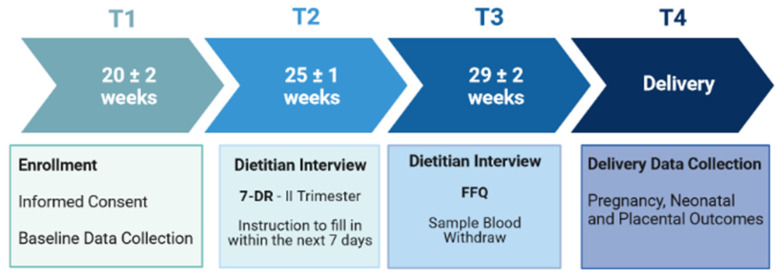
Study Design.

**Figure 2 nutrients-14-03631-f002:**
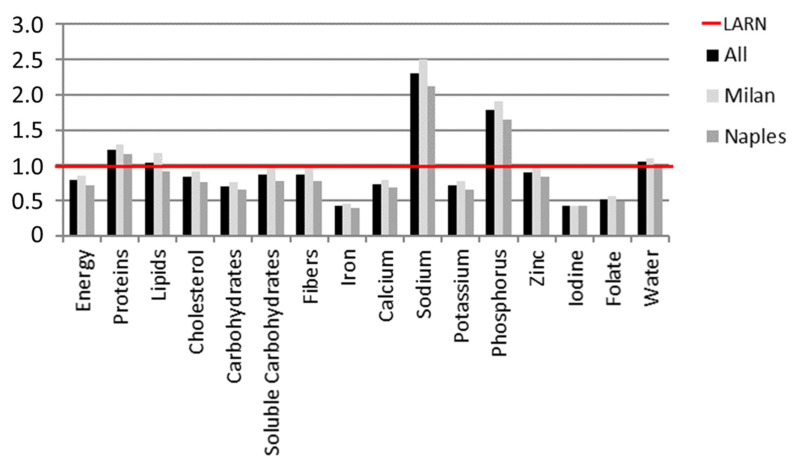
Energy and nutrients intakes referred to Italian LARN recommendations. Energy and nutrient intakes of the study population compared to the Italian LARN references. The average intake was calculated as bromatological data resulting from the analysis of the daily diaries weighed over 7 days (7-DR diaries).

**Table 1 nutrients-14-03631-t001:** Maternal characteristics and biomarker concentrations in the total study population and subgroups.

**Maternal Characteristics**		**TOTAL STUDY POPULATION** ***n* = 179**	**MILAN** ***n* = 85**	**NAPLES** ***n* = 94**	***p*-Value**
Maternal Age, years		31.8 ± 4.3	31.7 ± 4.5	31.9 ± 4.1	ns
Maternal pregestational BMI, kg/m^2^		21.9 ± 2.7	21.6 ± 2.8	22.3 ± 2.7	ns
Educational Level, %	low	18 (9.9%)	6 (7.0%)	12 (12.6%)	ns
	intermediate	55 (30.4%)	32 (37.2%)	23 (24.2%)	
	high	108 (59.7%)	48 (55.8%)	60 (63.2%)	
Working Status, %	unemployed	31 (17.4%)	5 (6.0%)	26 (27.4%)	<0.001
	worker	147 (82.6%)	78 (94.0%)	69 (72.6%)	
Marital Status, %	not married	64 (35.8%)	43 (50.6%)	21 (22.3%)	<0.001
	married	115 (64.2%)	42 (49.4%)	73 (77.7%)	
Supplement Use, %	none	19 (10.9%)	12 (14.6%)	7 (7.6%)	<0.001
	iron/folic acid	37 (21.3%)	5 (6.1%)	32 (34.8%)	
	multivitamin	118 (67.8%)	65 (79.3%)	53 (57.6%)	
**Nutritional, Inflammatory and** **Oxidative Biomarkers at T3**		**TOTAL STUDY POPULATION** **(*n* = 166)**	**MILAN** **(*n* = 81)**	**NAPLES** **(*n* = 85)**	***p*-Value**
RBCs Folate, ng/mL		595.2 ± 134.2	521.3 ± 123.5	633.0 ± 104.9	<0.001
Serum Vitamin D, ug/L		22.1 ± 7.6	21.2 ± 7.4	22.9 ± 7.7	ns
Plasma Hpn-25, ng/mL		4.4 ± 2.9	4.9 ± 3.0	4.0 ± 2.7	0.03
Plasma TAC, mM		2.8 ± 1.4	2.9 ± 1.5	2.7 ± 1.3	ns

Values are expressed as mean ± standard deviation. Data were analyzed according to their distribution with independent samples Student’s *t*-test or Mann–Whitney U test or Chi-Squared test for independence (with Yates continuity correction) or Fisher’s exact-test; statistical significance compared to Milan subgroup. Statistical significance: *p* < 0.05. BMI: Body Mass Index; RBCs: Red Blood Cells; Hpn-25: hepcidin mature form; TAC: Total Antioxidant Capacity.

**Table 2 nutrients-14-03631-t002:** Delivery outcomes in the total study population and subgroups.

Delivery Outcome at T4	TOTAL STUDY POPULATION *n* = 173	MILAN *n* = 83	NAPLES *n* = 90	*p*-Value
Maternal GWG, kg	12.6 ± 4.2	13.4 ± 4.3	12.0 ± 4.0	0.05
GA at Delivery, weeks	39.6 ± 1.2	39.8 ± 1.2	39.5 ± 1.2	0.02
Placental Weight, gr	541.4 ± 104.2	541.5 ± 110.5	542.4 ± 99.8	ns
N/P weight ratio	6.1 ± 1.0	6.1 ± 1.1	6.1 ± 0.9	ns
Neonatal Weight, gr	3231.7 ± 454.23	3232.7 ± 503.9	3225.2 ± 406.2	ns
NPI, gr/cm^3^	2.7 ± 0.3	2.7 ± 0.3	2.7 ± 0.3	ns
Neonatal Length, cm	49.2 ± 2.1	49.4 ± 2.3	49.1 ± 1.9	ns
Neonatal HC, cm	34.3 ± 1.3	34.1 ± 1.3	34.5 ± 1.3	0.02

Values are expressed as mean ± standard deviation. Data were analyzed according to their distribution with independent samples Student’s *t*-test or Mann–Whitney U test; statistical significance compared to Milan group. Statistical significance: *p* < 0.05. GWG: Gestational Weight Gain according to IOM guidelines; GA: Gestational Age; N: Neonatal; P: Placental; NPI: Neonatal Ponderal Index: {[neonatal weight (gr)/neonatal length (cm)]^3^} × 100 [33]; HC: Head Circumference.

**Table 3 nutrients-14-03631-t003:** Dietary patterns extraction from FFQ analysis: relation between food groups and dietary patterns expressed by factor loadings.

	‘HIGH MEAT, ANIMAL FATS, GRAIN’ DP	‘HIGH FISH, FRUIT, NUTS’ DP	‘HIGH EGG and SWEETS, LOW LEGUMES’ DP
**EXPLAINED VARIANCE, %**	11.7	11.4	10.3
**DAIRY**	0.057	0.508	−0.050
**CEREALS**	0.548	−0.210	−0.324
**GRAINS**	**0.625**	−0.027	−0.207
**VEGETABLES**	0.107	0.448	−0.280
**LEGUMES**	0.286	0.022	**−0.414**
**POTATOES**	0.563	−0.113	0.114
**MEAT**	**0.575**	−0.023	−0.255
**FISH**	0.177	**0.718**	−0.210
**EGG**	0.164	0.356	**0.515**
**FRUIT**	0.151	**0.520**	0.112
**NUTS**	0.053	**0.531**	0.261
**VEGETABLE FATS**	0.038	0.362	−0.098
**ANIMAL FATS**	**0.566**	−0.170	0.399
**SWEETS**	0.334	−0.013	**0.489**
**NON-ALCOHOL BEVERAGE**	0.470	−0.169	0.164

The factor loadings indicate how much every food item correlates with the extracted dietary patterns. The factor loadings with the highest absolute value were used for labelling and highlighted in bold type. DP: dietary pattern.

**Table 4 nutrients-14-03631-t004:** Multivariate analysis of the associations between maternal dietary patterns, biomarkers and delivery outcomes in the total study population.

			β (95% CI)
**‘HIGH MEAT, ANIMAL FATS, GRAINS’ DP**	Vitamin D, ug/L		−3.9 (−6.9; −0.9) *
	Hpn-25, ng/mL		0.3 (0.0; 0.5) *
	GA, weeks	F	−0.5 (−0.9; −0.0) *
		M	−0.3 (−0.7; −0.2)
**Hpn-25**	Placental Weight, gr		−8.3 (−15.7; −0.8) *
**Vitamin D**	Neonatal Weight, gr		31.1 (9.2; 53.10) **

The model includes full adjustment for potential confounders (maternal age, pregestational BMI, education, working status, parity, vitamin supplement use, geographical area). The dietary pattern analysis includes the adjustment for energy intake. The biomarker analysis includes adjustment for gestational age at T3. The analysis on delivery outcomes includes further adjustment for gestational age at delivery and GWG at term. Effect estimates indicates the change amount in the dependent variable for every unit increase in the independent one. Statistical significance: * *p* < 0.05, ** *p* < 0.01. CI: confidence interval.

## Data Availability

All data that support the findings of this study are available from the authors G.MA and F.P. on reasonable request.

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
