# Peer review of "Associations between Maternal Dietary Patterns, Biomarkers and Delivery Outcomes in Healthy Singleton Pregnancies: Multicenter Italian GIFt Study"

_nutrients, 2022, doi:10.3390/nu14173631_

Round 1
Reviewer 1 Report
1) The concept of ‘pattern adherence’ is not that clear to the reader. Similarly, the phrase ‘factor loadings’ in the legend of table 3 is somewhat confusing. I think it would be exceedingly useful have more explanation about the frequency analysis and how the analysis is interpreted within the results or method section. It is unclear if ‘component’ (as used in lines 320-326) indicates parts of the diet overall or proportion of women within each group.
Also, while table 4 presents the data it is not the easiest to judge how meaningful any of it is. For example, how important is a half week earlier gestational age for female babies whose mothers fell into the ‘high meat, animal fats, grains’ DP? Is half a week relevant if they were considered term gestations (ie weeks 38-42)?
I guess what is not clear is something like ‘maternal adherence to the ‘high meat, animal fats, grains’ dietary pattern was positively associated with Hpn-25 concentrations ……… the same dietary pattern showed a negative association with gestational age at delivery….’. Is each woman given a value of her adherence to a dietary pattern?
There seems to be some interesting data here, but the reporting of the dietary pattern adherence, or perhaps just the definition of the terms, is unclear to a reader who is not familiar with this analysis. A careful re-write of the section from lines 311-349 is required to make this more easily comprehensible.
2) There is a lot of examples of hyperbole language for what is meant to be a scientific report. Examples: line 306 and line 476 ‘Alarming’, line 368 ‘seem more and more relevant and alarming’. These statements are somewhat excessive and should be removed. Also, the phrase ‘the disappointing data’ line 424 seems somewhat emotive for a scientific report and should be removed.
3) In the limitations there should be some discussion about how when people know their food intake is being assessed can influence how much they eat. Is the lower-than-expected energy intake due in part to people’s choices being influenced by the fact that they are recording their food consumption? How reliable is self-reporting of food intake?
4) It might be helpful to include the reference range for vitamin D in pregnancy. Lines 418-420 mention the negative effects of severe vitamin D deficiency, but there is no indication how the values in the current study relate to this severe vitamin D deficiency. Also, there should be a citation for this statement describing the adverse outcomes of severe vitamin D deficiency.
5) Table 2
It is mathematically impossible for the average total study population for placental weight to be lower than the individual averages of the Milan cohort and the Naples cohort. Perhaps there is a typo in one of these numbers and this should be re-checked.
Minor
1) Line 133 ‘die’ should be ‘day’
2) line 67 ‘correct wrong exposures’, this is perhaps not the best phrase, I suggest changing it to ‘to evaluate deficiencies and ensure optimal exposure’
3) line 283 and 440 ‘gender’ should be ‘sex’. The gender identity of a baby is unable to be assessed and it is likely that what has been assessed here is the biological sex of the baby based on genitalia hence ‘sex’ not ‘gender’ is correct.
Author Response
Comments and Suggestions for Authors
1) The concept of ‘pattern adherence’ is not that clear to the reader. Similarly, the phrase ‘factor loadings’ in the legend of table 3 is somewhat confusing. I think it would be exceedingly useful have more explanation about the frequency analysis and how the analysis is interpreted within the results or method section. It is unclear if ‘component’ (as used in lines 320-326) indicates parts of the diet overall or proportion of women within each group.
- We thank the reviewer for this comment, agreeing that the Principal Component Analysis (PCA) for dietary pattern extraction may be not immediately understandable by readers not involved in nutritional studies. ‘Component’ indicates the overall dietary pattern as reported between as given in brackets (i.e.). The manuscript has been modified accordingly, and a more detailed description has been included in the ‘Statistical Analysis’ section (lines 233-247).
Also, while table 4 presents the data it is not the easiest to judge how meaningful any of it is. For example, how important is a half week earlier gestational age for female babies whose mothers fell into the ‘high meat, animal fats, grains’ DP? Is half a week relevant if they were considered term gestations (ie weeks 38-42)?
- We completely understand the reviewer’s point of view, therefore including a sentence in the discussion according to this comment. We agree that the clinical meaning of the results is questionable, but it needs to the related to the very strict inclusion criteria. These results were obtained from healthy singleton normal weight uncomplicated pregnancies and, despite being statistically significant, we detected a small difference in gestational age at delivery in association with the adherence to the ‘high meat, animal fats, grains’ DP. It is possible that by extending the inclusion criteria to high-risk pregnancies (e.g. obesity) the detected effect might be more clinically relevant. Moreover, recent animal studies reported the first evidences of sexual dimorphism on early exposure to maternal high fat diet during perinatal period. [PMID: 33914222; PMID: 34997206]. We now discuss these concepts from lines 472-485.
I guess what is not clear is something like ‘maternal adherence to the ‘high meat, animal fats, grains’ dietary pattern was positively associated with Hpn-25 concentrations the same dietary pattern showed a negative association with gestational age at delivery’. Is each woman given a value of her adherence to a dietary pattern? There seems to be some interesting data here, but the reporting of the dietary pattern adherence, or perhaps just the definition of the terms, is unclear to a reader who is not familiar with this analysis. A careful re-write of the section from lines 311-349 is required to make this more easily comprehensible.
-We thank the reviewer for this suggestion and the performed analysis has been explained more in detail, according to the first suggestion (lines 233-247).
2) There is a lot of examples of hyperbole language for what is meant to be a scientific report. Examples: line 306 and line 476 ‘Alarming’, line 368 ‘seem more and more relevant and alarming’. These statements are somewhat excessive and should be removed. Also, the phrase ‘the disappointing data’ line 424 seems somewhat emotive for a scientific report and should be removed.
We followed the Reviewer suggestions by erasing all the examples of hyperbole language through the manuscript.
3) In the limitations there should be some discussion about how when people know their food intake is being assessed can influence how much they eat. Is the lower-than-expected energy intake due in part to people’s choices being influenced by the fact that they are recording their food consumption? How reliable is self-reporting of food intake?
We completely agree with the reviewer and according to this we had included a specific sentence in the Limitations Section (‘we cannot exclude that subsequent maternal modifications of nutritional habits towards a healthier behavior realized. In fact, repeated investigations of nutritional habits may create individual consciousness, despite no nutritional counselling or intervention was performed’- lines 517-523). What we meant is that, despite the dieticians did not perform any counselling or nutritional intervention, we cannot exclude that by only repeating 2 nutritional queries (7-DRs and FFQs) at different gestational week the woman may change her habits toward a healthier nutritional habit, thus representing a potential limitation to the study as mentioned. The sentence has been modified in order to clarify the concept. Anyway, we tried to avoid further bias by employing validated tools (7-DR and FFQ) performed by professional dietitians who trained adequately the survey participants on describing the foods, amounts consumed, cooking methods, etc. At the end of the recording period, the record was thoroughly reviewed with the subject [PMID: 12082515] as now specify at lines 142-145 of paragraph ‘Nutritional Data Collection’.
4) It might be helpful to include the reference range for vitamin D in pregnancy. Lines 418-420 mention the negative effects of severe vitamin D deficiency, but there is no indication how the values in the current study relate to this severe vitamin D deficiency. Also, there should be a citation for this statement describing the adverse outcomes of severe vitamin D deficiency.
We followed the Reviewer suggestion and added the reference range for vitamin D severe deficiency, plus two new citations [41, 42] for the statement above-mentioned (lines 455-457).
5) Table 2. It is mathematically impossible for the average total study population for placental weight to be lower than the individual averages of the Milan cohort and the Naples cohort. Perhaps there is a typo in one of these numbers and this should be re-checked.
We thank the reviewer for the precious suggestion, that prevent to report a wrong data in our table 2. We now correct the mistake.
Minor
1) Line 113 ‘die’ should be ‘day’
We correct the expression ‘die’ in ‘day’ at line 113 thus reporting ‘smoking habit (up to 5 cigarettes per day)’.
2) line 67 ‘correct wrong exposures’, this is perhaps not the best phrase, I suggest changing it to ‘to evaluate deficiencies and ensure optimal exposure’
We improve this sentence as suggested by the reviewer.
3) line 283 and 440 ‘gender’ should be ‘sex’. The gender identity of a baby is unable to be assessed and it is likely that what has been assessed here is the biological sex of the baby based on genitalia hence ‘sex’ not ‘gender’ is correct.
Thank you for the suggestion. We revise the mistake by changing ‘gender’ in ‘sex’.

Reviewer 2 Report
This manuscript describes the associations between maternal dietary habits, biomarker status, and pregnancy outcomes among Italian healthy normal-weight pregnancies. While the results are interesting, this reviewer has the following comments.
1. In Introduction, it mentions no decisive data are available on the effect of nutritional interventions among low-risk pregnancies. However, there did not seem to be any intervention in this study. Then, how does this study address the identified gap in the literature?
2. Please describe why those biomarkers were selected to be studied.
3. Please address the accuracy of 7-DR in accessing energy and nutrient intakes. Self-reported dietary record is notoriously known as inaccurate.
4. 7-DR and FFQ were collected at T2 and T3. Please clarify data at which time point were used in the analysis. The authors stated the FFQ was used to examine maternal dietary intake and pattern adherence over the previous three months.
5. Clarify how PAL level was decided.
6. Is maternal age reported anywhere?
7. Seasonal data on vitamin D was reported in results. Are these from a subgroup of women whose blood sample was collected in autumn and another subgroup in summer, and winter? Please describe how these data were generated, sample size, and if those support the conclusion.
Author Response
Comments and Suggestions for Authors
This manuscript describes the associations between maternal dietary habits, biomarker status, and pregnancy outcomes among Italian healthy normal-weight pregnancies. While the results are interesting, this reviewer has the following comments.
- In Introduction, it mentions no decisive data are available on the effect of nutritional interventions among low-risk pregnancies. However, there did not seem to be any intervention in this study. Then, how does this study address the identified gap in the literature?
We agree with the Reviewer. Our study did not have among its aims a dietary intervention. Therefore, we erased the misleading term ‘interventions’, being replaced with nutritional ‘counseling’. The concept about ‘no decisive data are available’ is truly related also to nutritional counselling among low-risk pregnancies (lines 74-76).
- Please describe why those biomarkers were selected to be studied.
We decided to compared maternal serum vitamin D between a continental (Milan) and a Mediterranean (Naples) cities searching to differences related to the geographical origin. Evermore evidences of hypovitaminosis are reported in the Mediterranean area (lines 455-457).
We chose hepcidin as biomarker for its double role as an enhancer of inflammation, and a negative regulator iron homeostasis (lines 438-440). Indeed, iron bioavailability in pregnancy is controlled by hepcidin. When body iron levels are depleted due to anemia or hypoxia, hepcidin expression is reduced, allowing for increased dietary iron absorption and mobilization.
Folate are critical for several biochemical reactions (e.g. amino acids production, DNA synthesis and methylation, homocysteine metabolism and NO production). A folate deficiency causes an increase of inflammatory cytokines’ release, oxidative stress and apoptosis. The growing adherence to a ‘Western resembling’ diet makes difficult the correct intake of food-derived folate from fruits, vegetables, cereals, thus increasing the risk of maternal hyperhomocysteinemia and of neonatal spina bifida or neural tube defects (inserted at lines 415-421).
Maternal adherence to the ‘Western resembling’ dietary pattern and sedentary lifestyles have been increasingly associated to higher levels of oxidative stress. The human antioxidant system includes enzymes or macro- (e.g. albumin, ceruloplasmin) or small- (e.g. ascorbic acid, α-tocopherol, β-carotene, uric acid and bilirubin) molecules. These molecules are both endogenous or food-derived antioxidants. The here-used assay detects the combined antioxidant activities of all its constituents (endogenous or food-derived) not separating the aqueous and the lipid-soluble ones. This provides a more complete biological information of the overall antioxidant capacity.
- Please address the accuracy of 7-DR in accessing energy and nutrient intakes. Self-reported dietary record is notoriously known as inaccurate.
We agree with the Reviewer that self-reported dietary record may be influenced by several factors, being sometimes inaccurate, as discussed in the ‘Strengths and Limitations’ paragraph (lines 517-523). Our dietitians were aware of the bias that intrinsically characterize 7-DR [PMID: 1952811], being updated with the general principles for the collection of national food consumption data [https://efsa.onlinelibrary.wiley.com/doi/epdf/10.2903/j.efsa.2009.1435].
Anyway, we tried to avoid further bias by employing a validated 7-DR performed by professional dietitians, who trained adequately the survey participants on describing the foods, amounts consumed, cooking methods, etc. At the end of the recording period, the record was thoroughly reviewed with the subject [PMID: 12082515]. This is now specified at lines 142-145 of paragraph ‘Nutritional Data Collection’.
- 7-DR and FFQ were collected at T2 and T3. Please clarify data at which time point were used in the analysis. The authors stated the FFQ was used to examine maternal dietary intake and pattern adherence over the previous three months.
We thank the Reviewer for this observation. We added further details about 7-DR (lines 138, 140-141) and FFQ (lines 170-173) collection and data recording.
7-DR were presented at T2 (25 ± 1 gw) to be fulfilled by all the enrolled women in the 7 consecutive days (until 26 ± 1 gw). They thereby recoded nutritional data of the II trimester of pregnancy.
The FFQ we used to assess the nutritional consumption, and specifically dietary intakes and pattern adherence was validated and developed for Italian population. Given that FFQs concern the previous three months before their administration at T3 (29 ± 1 gw), their data regard the II trimester as well.
We have modified Figure 1 to make time points clearer.
- Clarify how PAL level was decided.
As mentioned in the ‘Strengths and Limitations’ paragraph, no information on the physical activity intended as Physical Activity Level (PAL) was recorded in our multicenter prospective cohort. Lacking of the PAL of each patient, we chose the mean value among the reference values proposed by LARN as ‘reference value’, though excluding the highest physical activity factor (2.1) unlikely to be encountered in pregnancy.
We rephrased the concept of Physical Activity Level (PAL) by adding further details: ‘Concerning Physical Activity Level (PAL) we considered the 3 main levels described by LARN (1.45 for sedentary activity, 1.6 for moderate activity and 1.75 for elevated activity) as well as their average, excluding the highest PAL of 2.1, which was inconsistent with our population possible PAL’ (lines 159-164).
- Is maternal age reported anywhere?
We now added ‘maternal age’ in Table 1 as suggested.
- Seasonal data on vitamin D was reported in results. Are these from a subgroup of women whose blood sample was collected in autumn and another subgroup in summer, and winter? Please describe how these data were generated, sample size, and if those support the conclusion.
Seasonal data on vitamin D were evaluated in a total of 166 subjects, n= 81 from Milan and n= 85 from Naples, according to the size data reported in Table 1; we specified that at lines 283-286. Basically, when collecting the serum samples, our clinicians recorded also the month of collection. So that it was possible to derive the seasonality data for both Milan and Naples group within the statistical analysis performed by the Kruskal–Wallis test and post hoc analyses by Mann–Whitney U test (with Bonferroni correction for multiple groups: p ≤ 0.017).
